# IMAGE2POINT: 3D POINT-CLOUD UNDERSTANDING WITH 2D IMAGE PRETRAINED MODELS

## ABSTRACT

3D point-clouds and 2D images are different visual representations of the physical world. While human vision can understand both representations, computer vision models designed for 2D image and 3D point-cloud understanding are quite different. Our paper explores the potential of transferring 2D model architectures and weights to understand 3D point-clouds, by empirically investigating the feasibility of the transfer, the benefits of the transfer, and shedding light on why the transfer works. We discover that we can indeed use the same architecture and pretrained weights of a neural net model to understand both images and point-clouds. Specifically, we transfer the image-pretrained model to a point-cloud model by *inflating* 2D convolutional filters to 3D convolutional filters and **f**inetuning the inflated **i**mage-**p**retrained models (FIP). We find that models with minimal finetuning efforts — only on input, output, and optionally, batch normalization layers — can achieve competitive performance on 3D point-cloud classification, beating a wide range of point-cloud models that adopt task-specific architectures and use a variety of tricks. When finetuning the whole model, the performance gets further improved. Meanwhile, FIP improves data efficiency, reaching up to 10.0 points top-1 accuracy gain on few-shot classification. It also speeds up training of point-cloud models by up to 11.1x for a target accuracy.

## 1 INTRODUCTION

Point-cloud is an important visual representation for 3D computer vision. It is widely used in applications such as autonomous driving (Behley et al., 2019; Caesar et al., 2020; Yue et al., 2018), robotics (Armeni et al., 2017; Pomerleau et al., 2015; Xu et al., 2021), augmented and virtual reality (Sketchup, 2021; Wu et al., 2015; Shi et al., 2015), *etc.* A point-cloud represents visual information in a highly different way from a 2D image. A point-cloud consists of a set of unordered points lying on the object's surface, with each point encoding its spatial x, y, z coordinates and potentially other features. In contrast, a 2D image organizes visual features as a dense 2D pixel array. Due to the representation differences, 2D image and 3D point-cloud understanding are treated as separate problems. Image models and point-cloud models are designed to have different architectures and are trained on different types of data. Few research efforts have tried to directly transfer models from images to point-clouds or vice versa.

Intuitively, both 3D point-clouds and 2D images are visual representations of the physical world. Their low-level representations are drastically different, but they can represent the same underlying visual concept. Furthermore, human vision has no problem understanding both representations. However, can computer vision models trained on one modality understand the other?

Somewhat remarkably, the answer to the question above is: Yes. 2D image models trained on image datasets can be transferred to understand 3D point-clouds with minimal efforts. As illustrated in Figure 1, we transfer a 2D ConvNet to a 3D ConvNet whose input is a 3D voxel representation converted from a point-cloud. Based on a pretrained 2D ConvNet, we *inflate* its 2D convolutional filters to 3D by copying the filter weights along a third dimension. We add linear input and output layers to the network; and on a target point-cloud dataset, we only finetune the input/output layers, and optionally, the normalization layers, while keeping the pretrained model weights untouched. We term such partially-finetuned-image-pretrained models as *FIP-IO* (finetuning only input and output layers) or *FIP-IO+BN* (finetuning input, output, and BN layers). FIP-IO+BN can achieve competitive

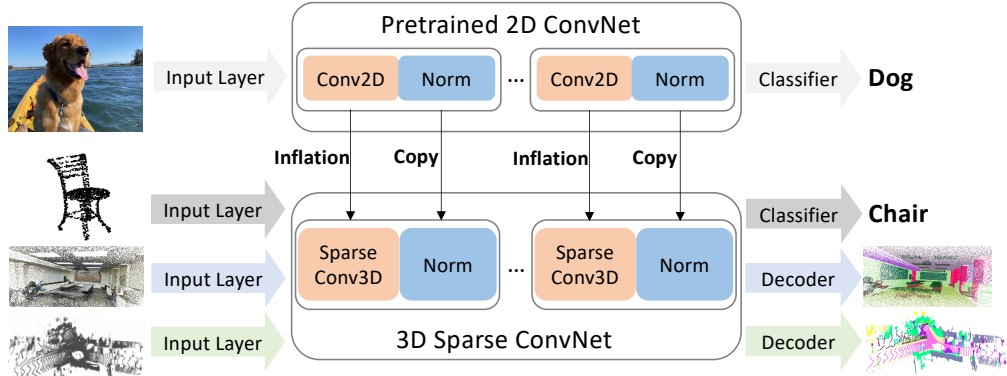

Figure 1: We investigate the feasibility of pretrained 2D ConvNets transferring to 3D sparse ConvNets. With filter inflation and finetuning only the input, output layer (classifier for classification task and decoder for semantic segmentation task), and optionally, normalization layers, 3D Sparse ConvNets are capable of dealing with point-cloud classification, indoor, and driving scene segmentation.

performance up to 90.8% top-1 accuracy on the ModelNet 3D Warehouse dataset, on top a ResNet50, outperforming previous point-cloud models that adopt task-specific model architectures and tricks.

Most point-cloud models except projection-based models are only trained from scratch. Based on our discovery, we further investigate fully-finetuned-image-pretrained models (termed as *FIP-ALL*). We observe that FIP-ALL brings significant improvement on top of ResNet series. Besides applying FIP-ALL to voxel-based method, we also find that it generalizes to other popular methods, such as point-based method (PointNet++ (Qi et al., 2017)) and projection-based method (SimpleView (Goyal et al., 2021a)), as well as current popular vision transformers (ViT (Dosovitskiy et al., 2020)). Specifically, FIP-ALL largely outperforms the training-from-scratch by 0.88, 0.50, 3.50, 4.18 points top-1 accuracy on top of PointNet++, SimpleView, ViT-B-16, and ViT-L-16, respectively. In addition to the performance gain, FIP-ALL exhibits superior data efficiency with up to 10.0 points improvement in few-shot classification on the ModelNet 3D Warehouse dataset. Comparing with training-from-scratch, FIP-ALL also dramatically speeds up the training by using 11.1 times fewer epochs to reach a target validation accuracy.

In order to understand why the image pretraining can be utilized to benefit point-cloud understanding, we conduct experiment studying the network dissection (Bau et al., 2017), text-shape representation transferring (Geirhos et al., 2018), and distribution distance.

## 2 RELATED WORK

### 2.1 POINT-CLOUD PROCESSING MODEL

**3D convolution-based method** is one of the mainstream point-cloud processing approaches which efficiently process point-clouds based on voxelization. In this approach, voxelization is used to rasterize point-clouds into regular grids (called voxels) so that conventional 3D convolutions can be applied. Sparse convolution is proposed to apply on the non-empty voxels (Liu et al., 2015; Choy et al., 2019; Tang et al., 2020; Zhou et al., 2020; Yan et al., 2018; Feng et al., 2021), largely improving the efficiency of 3D convolutions.

**Projection-based method** attempts to project a 3D point-cloud to a 2D plane and uses 2D convolution to extract features (Wang et al., 2018; Wu et al., 2018; 2019; Xu et al., 2020; Su et al., 2015; Lawin et al., 2017; Boulch et al., 2017). Specifically, bird-eye-view projection (Yang et al., 2018; Lang et al., 2019) and spherical projection (Wu et al., 2018; 2019; Xu et al., 2020; Milioto et al., 2019) make great progress in outdoor point-cloud tasks.

**Point-based method** directly processes the point-cloud data. The most classic methods, PointNet (Qi et al., 2016) and PointNet++ (Qi et al., 2017), consume points by customized feature aggregation. Many works further develop advanced local-feature aggregation operators that mimic the convolution

to structure data (Xu et al., 2021; Li et al., 2018b; Hua et al., 2018; Liu et al., 2019; 2020; Wang et al., 2017; Li et al., 2018a; Komarichev et al., 2019).

## 2.2 PRETRAINING IN 2D AND 3D VISION

**Pretraining in 2D vision** has shown effectiveness under supervised (Dosovitskiy et al., 2020; Girshick et al., 2014), self-supervised (Jing & Tian, 2020; Goyal et al., 2021b), and unsupervised contrastive approach (He et al., 2020; Bachman et al., 2019; Chen et al., 2020a; Caron et al., 2020; Chen et al., 2020c; Hjelm et al., 2018). After pretraining on a large amount of data, a 2D model requires much fewer computational resources and data for finetuning to reach competitive performance on downstream tasks (Kataoka et al., 2020; Caron et al., 2019; Chen et al., 2020b; Henaff, 2020).

**Pretraining in 3D vision** has been studied similarly as pretraining in 2D vision: both self-supervised and contrastive pretraining (Xie et al., 2020) show promising results. Due to the lack of large, annotated point-cloud datasets, pretraining in 3D vision is motivated to achieve data efficiency (Xu & Lee, 2020). Recent works (Hou et al., 2020; Zhang et al., 2021b) consider pretraining methods with data efficiency in mind, for example, Contrastive Scene Contents which makes use of both point-level correspondences and spatial contexts.

## 2.3 CROSS-MODAL TRANSFER LEARNING

**Cross-modal transfer learning** attempts to take advantage of data from different modalities (Dai & Nießner, 2018; Liu et al., 2021b). For example, Liu et al. (2021a) proposed pixel-to-point knowledge transfer (PPKT) from 2D to 3D which uses aligned RGB and RGB-D images during pretraining. Our work does not rely on joint image-point-cloud pretraining. Instead, we directly transfer an image-pretrained model to point-cloud with the simplest pretraining-finetuning scheme.

Some of the previous works for video and medical images (Carreira & Zisserman, 2017; Shan et al., 2018) have adopted the method of simply extending a pretrained 2D convolutional filter along time or depth direction for transferring to 3D models. Between language and image modality, transfer learning with minimal finetuning also shows a competitive performance (Lu et al., 2021).

## 3 CONVERTING A 2D CONVNET TO A 3D CONVNET

In this paper, we primarily focus on the 3D sparse-convolution based method to process point-clouds because it can be extended to all point-cloud tasks. As discussed in 2.1, we consider a set of points where each point is represented by its 3D coordinates and additional features such as intensity and RGB. We then voxelize/quantize these points into voxels according to their 3D space coordinates, following Choy et al. (2019). A voxel's feature is inherited from the point that lies in the voxel. If there are multiple points in a voxel, then we average all points' feature and assign the mean to the voxel. If there is no point in the voxel, then we simply set the voxel's feature to 0. When using sparse convolution, we skip the computation on empty voxels.

Given a pretrained 2D ConvNet, we convert it to a 3D ConvNet that takes 3D voxels as input. The key element of this procedure is to convert 2D convolution filters to 3D, *i.e.* constructing 3D filters with the weights directly inherited from 2D filters. A 2D convolutional filter can be represented with a 4D tensor of shape $[M, N, K, K]$, representing output dimension, input dimension, and two spatial kernel sizes, respectively. A 3D convolutional filter has an extra dimension, and its shape is $[M, N, K, K, K]$. To better illustrate, we ignore the output and input dimensions and only consider a spatial slice of the 2D filter with shape $[K, K]$. The simplest way to convert this 2D filter to 3D is to copy the 2D filter and repeat it by $K$ times along a third dimension. This operation is the same as the *inflation* technique used by (Carreira & Zisserman, 2017) to initialize a video model with a pretrained 2D ConvNet.

Besides convolution, other operations such as downsampling, BN, nonlinear activation can be easily migrated to 3D. Our 3D model inherits the architecture of the original 2D ConvNet, but we also add a linear layer as the input layer and an output layer depending on the target task. For classification, we use a global average pooling layer followed by one fully connected layer to get the final prediction. For semantic segmentation, the output layer is a U-Net style decoder (Ronneberger et al., 2015). The architecture of the input/output layers is described in more detail in Appendix A.9.

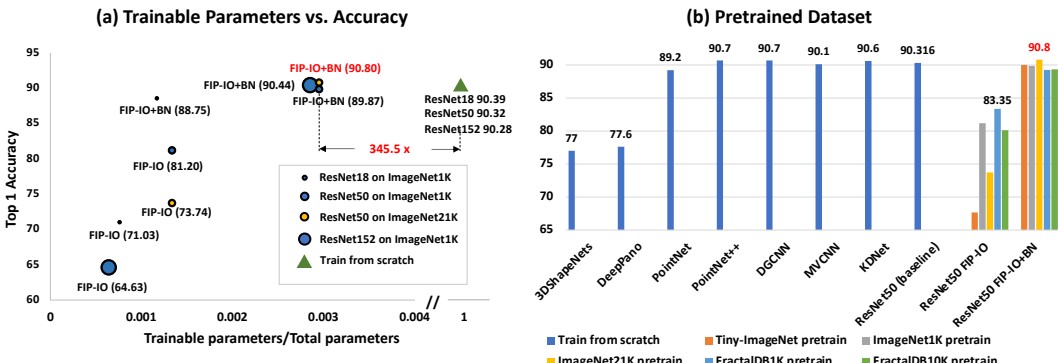

Figure 2: **a)** the left figure shows the trainable parameters ratio *w.r.t* top-1 accuracy on ModelNet 3D Warehouse dataset. **b)** the right figure shows the performance of FIP-IO and FIP-IO+BN on top of **ResNet50** pretrained on different datasets.

# 4 EMPIRICAL EVALUATION

To explore the image to point-cloud transfer, we study three settings: 1) partially-finetuned-image-pretrained model, only finetuning input and output layers (FIP-IO), 2) finetuning input, output, and batch normalization layers (FIP-IO+BN), and 3) finetuning the whole pretrained network (FIP-ALL). Under the three settings, we extensively explore the feasibility of transferring the image-pretrained model for point-cloud understanding and its benefits. The entire empirical evaluation is organized as four questions: 1) Can we transfer pretrained-image models to recognize point-clouds? (Section 4.1) 2) Can image-pretraining benefit the performance of point-cloud recognition? (Section 4.2) 3) Can image-pretrained model improve the data efficiency on point-cloud recognition? (Section 4.3) 4) Can image-pretrained model accelerate training point-cloud models? (Section 4.4)

**Datasets.** We benchmark the transferred models on ModelNet 3D Warehouse classification (Wu et al., 2015), S3DIS indoor segmentation (Armeni et al., 2017), and SemanticKITTI outdoor segmentation (Behley et al., 2019) tasks. ModelNet 3D Warehouse is a CAD model classification dataset that consists of point-clouds with 40 categories. CAD models in this benchmark come from 3D Warehouse (Sketchup, 2021). In this benchmark, we only utilize x, y, z coordinates as features. S3DIS is a dataset collected from real-world indoor scenes and includes 3D scans of Matterport Scanners from 6 areas. It provides point-wise annotations for indoor objects like chair, table, and bookshelf, *etc*. SemanticKITTI dataset from KITTI Vision Odometry (Geiger et al., 2012) is a driving scene dataset. It provides dense point-wise annotations for the complete 360 degrees field-of-view of the deployed automotive lidar, which is currently one of the most challenging datasets.

ResNet (He et al., 2016a) series is used mostly throughout our experiments. Depending on the experiments, ResNets are pretrained on Tiny-ImageNet, ImageNet-1K, ImageNet-21K (Deng et al., 2009), and Fractal database (FractalDB) (Kataoka et al., 2020). Our pretrained models are directly downloaded from various sources, with detailed links provided in Section A.1. To study the benefits of using pretrained image models, we also utilize PointNet++ (Qi et al., 2017), ViT (Dosovitskiy et al., 2020), and SimpleView (Goyal et al., 2021a) as our baselines.

## 4.1 CAN WE TRANSFER PRETRAINED-IMAGE MODELS TO RECOGNIZE POINT-CLOUDS?

To evaluate the feasibility of transferring pretrained 2D image models to 3D point-cloud tasks, we conduct experiments on top of the ResNet series since there are abundant open-source pretrained ResNet available. In particular, we convert 2D ConvNets into 3D ConvNets using the procedure described in Section 3. We hypothesize that, if a pretrained 2D image model is capable of understanding point-clouds directly, we can see a non-trivial performance by only finetuning input and output layers of the transferred model. Further, as we gradually relax the frozen parameters, finetuning BN parameters as well, the transferred model can achieve better performance, even surpassing training-from-scratch performance.

Table 1: ModelNet 3D Warehouse classification results (top-1 accuracy %) of fully-finetuned-image-pretrained models (FIP-ALL) based on different pretrained models. We include 2021 SOTAs, such as RSMix (Lee et al., 2021), Point Transformer (Point-Trans) (Zhao et al., 2021), DRNet (Qiu et al., 2021), and PointCutMix (Zhang et al., 2021a), for comparison.

| Method | ResNet18 | ResNet50 | ResNet152 | ResNet101×2 |
|---|---|---|---|---|
| From Scratch | 90.39 | 90.32 | 90.28 | 90.03 |
| FIP-ALL on ImageNet1K | 90.52 (**+0.13**) | 90.92 (**+0.60**) | 91.09 (**+0.81**) | 90.52 (**+0.49**) |
| FIP-ALL on ImageNet21K | - | 91.05 (**+0.73**) | - | - |

| Method | PointNet++(SSG) | ViT-B-16 | ViT-L-16 | SimpleView |
|---|---|---|---|---|
| From Scratch | 90.34 | 84.27 | 83.48 | 93.3 |
| FIP-ALL on ImageNet1K | 91.22 (**+0.88**) | - | - | 93.8 (**+0.50**) |
| FIP-ALL on ImageNet21K | - | 87.77 (**+3.50**) | 87.66 (**+4.18**) | - |

| Method | RSMix | Point-Trans | DRNet | PointCutMix |
|---|---|---|---|---|
| From Scratch | 93.5 | 93.7 | 93.1 | 93.4 |

Table 2: Indoor scene and outdoor scene segmentation results (mIoU %) of fully-finetuned-image-pretrained Model (FIP-ALL). In this table, all image-pretrained models are pretrained on ImageNet1K.

| Method | S3DIS (mIoU %) | | SemanticKITTI (mIoU %) | |
|---|---|---|---|---|
| | PointNet++(SSG) | ResNet18 | HRNetV2-W48 | ResNet18 |
| From Scratch | 52.45 | 55.09 | 44.12 | 64.75 |
| FIP-ALL on ImageNet1K | 55.01 (**+2.56**) | 56.62 (**+1.53**) | 47.53 (**+3.41**) | 65.57 (**+0.82**) |

We conduct two groups of experiments with FIP-IO and FIP-IO+BN, with the results shown in Figure 2. The first is to evaluate the performance as the trainable parameters gradually increase. As shown in Figure 2 (a), training **no more than 0.3 % (345.5x fewer)** of the whole parameters, the image pretraining even beats the training-from-scratch (100 % trainable parameters). Specifically, ResNet152 FIP-IO+BN with ImageNet1K pretraining improves training-from-scratch by 0.16 points, and ResNet50 FIP-IO+BN with ImageNet21K pretraining improves 0.48 points. Meanwhile, FIP-IO also reaches a non-trivial performance. ResNet50 FIP-IO pretrained on ImageNet1K achieves 81.20 % top-1 accuracy, only 9.12 points worse than training-from-scratch with approximately 0.1 % trainable parameters.

Furthermore, to investigate the effect of different datasets, as shown in the right figure of Figure 2, we inflate ResNet50 pretrained from different image datasets, including Tiny-ImageNet, ImageNet1K, ImageNet21K, FractalDB1K, and FractalDB10K, then evaluate on the ModelNet 3D Warehouse.

We discover that, even if we only finetune the input and output layers while keeping the image-pretrained weights frozen, the FIP-IO pretrained from ImageNet1K, FractalDB1K, and FractalDB10K achieves competitive performance. Specifically, ResNet50 FIP-IO with ImageNet1K pretraining outperforms 3D ShapeNet (Wu et al., 2015) and DeepPano (Shi et al., 2015), which were the state-of-the-arts in 2015, by 4.2 and 3.6 points respectively in top-1 accuracy on ModelNet 3D Warehouse. More importantly, with ImageNet21K pretrained model, ResNet50 FIP-IO+BN surpasses training-from-scratch by 0.48 points, even beating a variety of well-known methods including PointNet (Qi et al., 2016), MVCNN (Su et al., 2015), DGCNN (Wang et al., 2019), KDNet (Klokov & Lempitsky, 2017), *etc*.

Notably, we find out the answer to "Can we transfer pretrained-image models to recognize point-clouds?": Yes. The pretrained 2D image models can be directly used for recognizing point-clouds. It is also noteworthy that the pretraining dataset is not restricted to natural but also synthetic images like those in FractalDB1K/10K.

### 4.2 CAN IMAGE-PRETRAINING BENEFIT POINT-CLOUD RECOGNITION?

From the previous subsection, we find unexpectedly that image-pretrained model can be directly used for point-cloud understanding. In this subsection, we investigate whether image-pretrained model

Table 3: Comparison with PointContrast (Xie et al., 2020) on the ModelNet 3D Warehouse. PointContrast provides two different pretrained models with using PointInfoNCE loss and Hardest Contrastive loss, respectively.

| From scratch | PointInfoNCE | Hardest Contrastive | ImageNet1K pretrain (**Ours**) |
|---|---|---|---|
| 89.95 | 90.24 (**+0.29**) | 90.15 (**+0.20**) | 90.88 (**+0.93**) |

is helpful to improve the performance on point-cloud tasks. We use different baselines, including voxelization-based method (simply ResNet), point-based method (PointNet++ (Qi et al., 2017)), projection-based method (SimpleView (Goyal et al., 2021a)), and current popular transformer-based method (ViT-B-16 and ViT-L-16 (Dosovitskiy et al., 2020)), and fully finetune them on three point-cloud datasets: classification on ModelNet 3D Warehouse, indoor scene segmentation on S3DIS, and outdoor scene segmentation on SemanticKITTI, as shown in Table 1 and Table 2.

For PointNet++, we use ImageNet1K to pretrain: we break each image into pixels and regard it as a point-cloud. For ViT, we directly use the open-source pretrained model and finetune it on ModelNet 3D Warehouse. All the implementation details are illustrated in Appendix A.1.

Table 1 presents performance on ModelNet 3D Warehouse dataset. We observe that FIP-ALL improves all baselines steadily and significantly. Besides, pretraining brings more improvements to deeper models. For example, ResNet18 can only be improved by 0.13% top-1 accuracy, but pretraining on ImageNet1K leads to 0.81 points top-1 accuracy improvement on top of ResNet152. Moreover, larger pretrained datasets also lead to better performance. Specifically, ResNet50 FIP-ALL from ImageNet21K can reach 91.05% top-1 acc, with 0.73 points improvement over training-from-scratch. Such FIP-ALL significantly outperforms a series of well-known methods such as (Qi et al., 2016; 2017; Klokov & Lempitsky, 2017; Wang et al., 2019; Su et al., 2015; Li et al., 2018a).

We also explore FIP-ALL on different architectures, as shown in the second group of Table 1. In particular, FIP-ALL on top of PointNet++, ViT-B-16, ViT-L-16 and SimpleView with image dataset pretraining improve the training-from-scratch by 0.88, 3.50, 4.18, 0.50 points, respectively. Especially for the current superior baseline in image recognition, ViT-B-16 and ViT-L-16, the improved performance is quite significant, revealing the huge potential of using image-pretrained models for point cloud recognition.

For the challenging indoor and outdoor scene segmentation, using ImageNet1K pretrained models (FIP-ALL on ImageNet1K) also improve the training-from-scratch consistently, as shown in Table 2. PointNet++ (resp. ResNet18) pretrained on ImageNet1K outperforms the training-from-scratch by 2.56 points (resp. 1.53 points) mIoU on S3DIS dataset. For SemanticKITTI, we utilize the commonly used projection-based method with 2D ConvNet HRNet. With ImageNet1K pretraining, we observe 3.41 points mIoU improvement, a large margin in such a challenging task. Since HRNetV2-W48 has rich pretrained models, we finetune Cityscapes pretrained HRNetV2-W48 and observe this enhances more (5.25% mIoU improvement over training from scratch). Even for the ResNet18 with a high from-scratch performance of 64.75% mIoU, the ImageNet1K pretraining can also bring 0.82 points mIoU improvement.

Finally, we compare the performance gain with the well-known point-cloud self-supervised method PointContrast (Xie et al., 2020), as presented in Table 3. We use the same model architecture and finetuning recipe, and the only difference is the pretraining weights. We can observe that image-pretraining on ImageNet1K significantly boosts the training-from-scratch by 0.93 points, surpassing the PointContrast by at least 0.64 points.

Therefore, the answer to "Can image-pretraining benefit point-cloud recognition" is: Yes. Image-pretraining can indeed improve point-cloud recognition, which can generalize to a wide range of backbones and benefit more challenging tasks.

### 4.3 CAN IMAGE-PRETRAINED MODELS IMPROVE THE DATA EFFICIENCY ON POINT-CLOUD RECOGNITION?

Data efficiency is extremely important in point-cloud understanding due to the huge labor of collecting and annotating point-cloud data. In this subsection, we investigate whether the image-pretrained

Table 4: Few-shot experiments on top of different ResNets on the ModelNet 3D Warehouse dataset.

| Few-shot | ResNet18 | ResNet50 (from scratch/FIP-ALL) | ResNet152 |
|---|---|---|---|
| 10-shot | 72.2±0.8/73.2±0.6 (**+1.0**) | 71.7±0.7/74.1±0.8 (**+2.4**) | 69.8±1.1/73.9±0.4 (**+4.1**) |
| 5-shot | 63.7±1.6/66.6±0.8 (**+2.9**) | 62.4±1.1/66.0±2.2 (**+3.6**) | 59.4±0.8/66.5±0.9 (**+7.1**) |
| 1-shot | 26.8±4.4/36.8±0.6 (**+10.0**) | 28.1±0.4/34.1±0.2 (**+6.0**) | 23.3±4.3/33.2±1.3 (**+9.9**) |

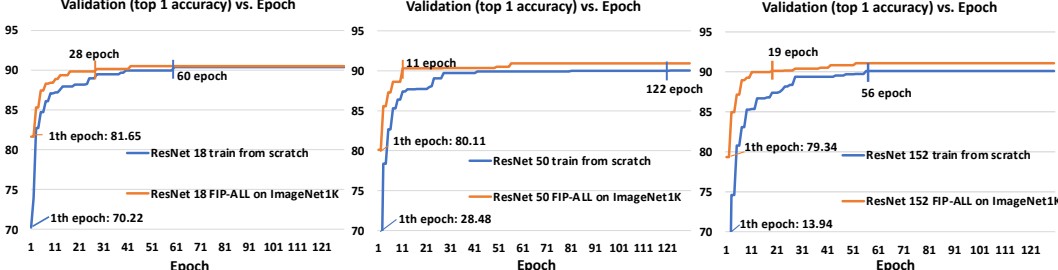

Figure 3: The curves of validation accuracy w.r.t training epoch. We compare the results between training-from-scratch and FIP-ALL on the ImageNet1K, on top of ResNet18, ResNet50, and ResNet152, respectively.

model can help to improve the data efficiency by conducting few-shot setting experiments, including 1-shot, 5-shot, and 10-shot. We conduct 3 trials for each setting and report the results as mean ± std.

In detail, for each class (ModelNet 3D Warehouse involves 40 classes), we randomly choose a few point-clouds as training data, and still evaluate on the whole test set. We compare the results between training-from-scratch and FIP-ALL pretrained on the ImageNet1K dataset. The experimental results are shown in Table 4. We observe that FIP-ALL dramatically surpasses training-from-scratch on the low data regime (1-shot): pretraining on ImageNet1K brings 10.0, 6.0, and 9.9 points top-1 accuracy improvement for ResNet18, ResNet50, and ResNet152, respectively. For 5-shot and 10-shot settings, using ImageNet1K pretraining can still consistently improve the performance. However, we also observe that as the amount of training data increases, the performance gain becomes saturated.

Therefore, our answer to "Can image-pretrained model improve the data efficiency on point-cloud recognition?" is: Yes. Image-pretrained model can improve the data efficiency on point-cloud recognition, especially on low data regime. When the training data increases, it can still improve the performance, but the gain becomes marginal.

## 4.4 CAN IMAGE-PRETRAINED MODELS ACCELERATE POINT-CLOUD TRAINING?

We also investigate whether image-pretrained model can help point-cloud task train faster. The results are shown in Figure 3.

We discover that, after training only one epoch on ModelNet 3D Warehouse dataset, FIP-ALL on ImageNet1K achieves very impressive performance, yet the performance of training-from-scratch is still at a low level. For example, after the first epoch, ResNet50 (resp. ResNet152) with training from scratch can only achieve 28.48% (resp. 13.94%) top-1 accuracy while ResNet50 (resp. ResNet152) with ImageNet1K pretraining reaches 80.11% (resp. 79.34%) top-1 accuracy. Moreover, to reach 90% top-1 accuracy, a non-trivial performance, FIP-ALL significantly accelerates the training by 2.14x (28 vs. 60 epoch), 11.1x (11 vs. 122 epoch), 2.95x (19 vs. 56 epoch) over training-from-scratch, on top of ResNet18, ResNet50, and ResNet152, respectively.

Therefore, our answer to "Can image-pretrained model accelerate point-cloud training?" is still: Yes. The image-pretrained model can significantly accelerate the training speed of point-cloud tasks.

Table 5: Texture-shape representation transferring experiment on ModelNet 3D Warehouse.

| Method | Pretrained dataset | top-1 accuracy |
|---|---|---|
| ResNet50 FIP-IO | ImageNet1K | 81.20 |
| ResNet50 FIP-IO | Stylized-ImageNet | 83.52 |
| BagNet17 FIP-IO | ImageNet1K | 57.53 |
| BagNet33 FIP-IO | ImageNet1K | 68.40 |

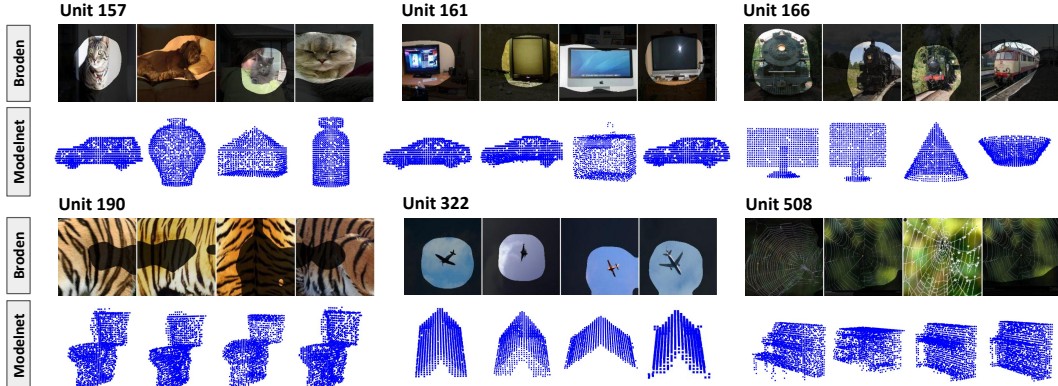

Figure 4: Network dissection of FIP-IO+BN. The visualization displays what the units are attending to on both image dataset and ModelNet 3D Warehouse dataset.

## 5 DISCUSSION

In this section, we attempt to shed light on why transferring image-pretrained models for point-cloud understanding works. Inspired by recent related works (Geirhos et al., 2018; Brendel & Bethge, 2019; He et al., 2015; 2016b; Bau et al., 2017), we explore this from the aspects of the network dissection, texture-shape representation transferring, and feature distribution distance.

### 5.1 WHAT DO IMAGE-PRETRAINED MODELS TRANSFER TO POINT-CLOUD MODEL?

**Does the image-pretrained model transfer the visual concepts?** Inspired by the network dissection for 2D Broden dataset (Bau et al., 2017), we also attempt to explore what the transferred units are focus on in the ModelNet 3D Warehouse dataset. We present the visualization of FIP-IO+BN pretrained on ImageNet1K. The visualization shows the most activated cases when the whole dataset passes through each unit of the last model stage, as displayed in Figure 4. More visualization can be found in Appendix 7. From the visualization alone, we do not get obvious cues of what visual concepts are transferred between the two modalities. For example, unit 161 strongly activating to computer screens in the Broden, yet it attends to cars and shelves in the ModelNet 3D Warehouse. Surprisingly, we find that the pretrained units are prone to cluster similar objects. In fact, such clustering ability is an important cue of performing well on classification tasks (Hartigan & Wong, 1979; Caron et al., 2021).

**Do pretrained-image models transfer shape or texture representation?** Recent work (Geirhos et al., 2018) proposed that models learn texture and shape from ImageNet. We follow this direction to explore what pretrained-image mode transfers. For the experiment, we take two image-pretrained models with either more shape or texture representation and compare their FIP-IO performance.

In order to force the pretrained models to acquire more shape representation, Geirhos et al. (2018) stylizes the images in ImageNet into artwork style, such that the models trained on those images are confused by variant textures, hence having stronger shape representation. We directly take the pretrained ResNet50 on stylized ImageNet as the stronger shape representation model.

To get a stronger texture representation model, we are inspired by BagNet (Brendel & Bethge, 2019). By controlling the receptive field, BagNet breaks the shape in an image and focuses more on the texture information. Our experimental result is shown in Table 5. BagNet17 means the size of attended patches is $17 \times 17$, and BagNet33 means the size of attended patches is $33 \times 33$. Note that after inflating the BagNet17 and BagNet33, both architectures are the same as inflated ResNet50. Besides, both BagNet17/33 and ResNet pretrained on stylized-ImageNet1K perform worse than the original ResNet50 on ImageNet (Geirhos et al., 2018; Brendel & Bethge, 2019).

We can observe that the ResNet50 FIP-IO on Stylized-ImageNet (with stronger shape representation) outperforms the baseline ResNet50 FIP-IO on ImageNet1K over 2.32 points top-1 accuracy, while both inflated BagNets perform dramatically worse than the baseline. This shows that the **shape representations** are transferred.

To make clear what parts of the image are critical for the transfer, we further conduct experiments using different image processing methods for pretraining. The results are reported in Appendix A.7.

### 5.2 WHY DOES FINETUNING BATCH NORMALIZATION HELP THE TRANSFERRING?

Our experiment in Figure 2 shows that finetuning BN, in addition to the input and output layer, can greatly improve the transfer performance. It is interesting why such a small part of the network, in terms of parameter size and FLOPs, can have a big impact.

We calculate the first-wasserstein distance (FWD) (Rubner et al., 2000), a measurement of Gaussian distribution distance, between the 2D image and 3D point-cloud feature distributions. For estimating the 2D feature distribution, we pass the whole ImageNet1K dataset into the pretrained-image models, collect the pre-activation features after each convolution layer, then sample 15,000 data points from the element-wise distribution of collected features (we assume the pre-activation features present Gaussian distribution). For 3D feature distribution, we use FIP-IO, FIP-IO+BN, and FIP-ALL to conduct the same operation on the ModelNet 3D Warehouse dataset and also collect 15,000 data points. We then calculate the FWD between the element-wise feature distribution of image-pretrained model and each of the FIP-IO, FIP-IO+BN, FIP-ALL model.

We observe that the layer-wise average FWD between FIP-IO (point-cloud features) and image model (image features) is $2.1 \times 10^2$ , yet after finetuning batch normalization layers, the distance is dramatically reduced. Particularly, average FWD between FIP-IO+BN and image model is $0.27$, and FWD between FIP-ALL and image model is only $0.093$. Complete FWD for each layer is shown in Appendix Table 11. This suggests that batch normalization plays a critical role in transforming the point-cloud representation to be closer to the image representation.

## 6 CONCLUSION

In this work, we use finetuned-image-pretrained models (FIP) to explore the feasibility of transferring image-pretrained models for point-cloud understanding and the benefits of using image-pretrained models on point-cloud tasks. We surprisingly discover that, with simply inflating a 2D pretrained ConvNet and minimal finetuning — input, output, and optionally, batch normalization layer (FIP-IO or FIP-IO+BN), FIP can achieve very competitive performance on 3D point-cloud classification, beating a wide range of point-cloud models that adopt a variety of tricks. Moreover, we find that when finetuning all the parameters of the pretrained models (FIP-ALL), the performance can be significantly improved on point-cloud classification, indoor and outdoor scene segmentation. Fully finetuned models generalizes to most of the popular point-cloud methods. We also find that FIP-ALL can improve the data efficiency on few-shot learning and accelerate the training speed by a large margin. Additionally, we shed light on why image-pretrained models can be used for point-cloud understanding from three aspects: network dissection, texture-shape representation transferring, and feature distribution distance. Compared with previous works that seek improvements from designing architectures and pretraining only on point-cloud modality, our work is not limited by the architecture design and the small-scale point-cloud dataset. We believe that image pretraining is one of the solutions to the bottleneck of point-cloud understanding and hope this direction can inspire the research community.

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

## A  APPENDIX

### A.1  IMPLEMENTATION DETAIL.

Our experiments are conducted on ModelNet 3D Warehouse, S3DIS, and SemanticKITTI datasets. For the ModelNet 3D Warehouse dataset, we train all models on the train set and evaluate on the validation set. For the S3DIS, we train all models on area 1, 2, 3, 4, 6 and evaluate on area 5. For the SemanticKITTI dataset, we train all models on splits 00-10 except 08 which is used for evaluation. For each of the dataset, all ResNet series models use the same training scheme, and all experiments are implemented with PyTorch.

When training on the *ModelNet 3D Warehouse dataset*, coordinates of point-clouds are randomly scaled, translated, and jittered. We use SGD optimizer with momentum 0.9, weight-decay $10^{-4}$, and initial learning rate 0.1 with cosine learning rate scheduler. Each mini batch is set to 32, and models are trained for 300 epochs. For both training and inference phase, we only utilize x, y, z coordinates without other features and set voxel size to 0.05. The experiments for ModelNet 3D Warehouse are all conducted on a Titan RTX GPU.

When training on the *S3DIS dataset*, we concatenate all subparts of an indoor scene to train and validate on. Along x, y directions, scenes are applied horizontal flip randomly. RGB features are randomly jittered, translated, and auto contrasted. Finally, we normalize and clip point-clouds. We set voxel size to 0.05, use SGD optimizer with momentum 0.9, weight-decay $10^{-4}$, and initialize learning rate to 0.1 with polynomial learning rate scheduler. Each mini batch is set to 3, and models are trained for 400 epochs on 2 Titan RTX GPUs.

When training on the *SemanticKITTI dataset*, coordinates of each point-cloud are randomly scaled and rotated. We use SGD optimizer with momentum 0.9, weight-decay $10^{-4}$, and initial learning rate 0.24 with cosine warmup learning rate scheduler. Each mini batch is set to 2, and models are trained for 15 epochs on 4 Titan RTX GPUs. For both training and inference phase, we utilize x, y, z coordinates as well as intensity feature and set voxel size to 0.05.

Most of our pretrained models come from open-sources, [1] [2] [3] [4] [5] [6] , so we do not need to take time and computational resources for pretraining. We use torchsparse [7] to produce sparse 3D convolutions.

**Details of Section 4.1.** In this section, we take the ResNet architecture, inflate the pretrained models of different image datasets, and add linear input and output layers as shown in Section A.9. The ResNet50 pretrained on ImageNet1K is directly taken from PyTorch. We use the same training recipe provided by PyTorch to train the ResNet50 on Tiny-ImageNet. The pretrained ResNet50 on ImageNet21K comes from Ridnik et al. (2021).

**Details of Section 4.2, Section 4.3 and 4.4.** In this section, the ResNet pretrained models are taken from the same sources as illustrated above.

For PointNet++ pretraining on ImageNet1K, we utilize the PointNet++ SSG version (Qi et al., 2017). We break the image into pixels and regard the group of pixels as a point-cloud with coordinates of x, y positions in the original image and appending $z = 1$ to all pixels. Then, we set center sampling number to 1024 and 256 for first and second stage, and the radius is set into 8 and 64, respectively. For each center point, we query 64 neighbouring points. The training recipe is also provided by PyTorch.

For ViT models, we directly take the pretrained weights from Dosovitskiy et al. (2020). To apply it on ModelNet 3D Warehouse, we sample 256 centers and group 64 nearby points, regarding these as "point-cloud patches". Then, we use a linear embedding to project the point-cloud patches into a sequence, and ViT processes them same as image patches. Except the linear embedding and the final

---

[1]https://pytorch.org/vision/stable/models.html

[2]https://github.com/Alibaba-MIIL/ImageNet21K

[3]https://github.com/hirokatsukataoka16/FractalDB-Pretrained-ResNet-PyTorch

[4]https://github.com/HRNet/HRNet-Semantic-Segmentation/tree/pytorch-v1.1

[5]https://github.com/rgeirhos/Stylized-ImageNet

[6]https://github.com/wielandbrendel/bag-of-local-features-models

[7]https://github.com/mit-han-lab/torchsparse

Table 6: ResNet50 results (evaluated on ModelNet 3D Warehouse) of fintuning the mean and variance in batch normalization layers on different datasets. IO indicates finetuning input and output layer, IOms indicates updating input, output, mean and variance, IOmsWb indicates finetuning input, output and the whole BN.

| Layers | Tiny-ImageNet | ImageNet1K | ImageNet21K | FractalDB1K | FractalDB10K |
|--------|---------------|------------|-------------|-------------|--------------|
| IO | 67.666 | 81.199 | 73.744 | 83.347 | 80.105 |
| IOms | 83.793 | 82.942 | 84.076 | 72.326 | 79.66 |
| IOmsWb | 89.992 | 89.87 | 90.721 | 89.263 | 89.344 |
| From scratch | 90.316 | 90.316 | 90.316 | 90.316 | 90.316 |

Table 7: ResNet18, 50, 152 results (evaluated on ModelNet 3D Warehouse) of finetuing the mean and variance in batch normalization layers.

| Layers | ResNet18 | ResNet50 | ResNet152 |
|--------|----------|----------|-----------|
| IO | 71.029 | 81.199 | 64.627 |
| IOmv | 81.888 | 82.942 | 82.658 |
| IOmvWb | 88.574 | 89.87 | 90.438 |
| From Scratch | 90.397 | 90.316 | 90.276 |

output classifier, all the models are kept same as the origin version. For the experiments on S3DIS and SemanticKITTI, the architecture detail of ResNet18 is shown in A.5 listing 2.

For SimpleView model, all the experiment settings are the same as Goyal et al. (2021a). The only difference is whether to use the pretrained ResNet18. For HRNetV2-W48, we directly use the ImageNet1K and Cityscape pretrained models from (Sun et al., 2019).

We conduct three trials on the few-shot experiments. For each trial, we change the random seed but keep all the other settings the same. To plot the training speed curve, we directly use the training log without any other changes, such as smoothing.

## A.2 FINETUNING THE MEAN AND VARIANCE OF BATCH NORMALIZATION.

For the first group of experiments, ResNet50 FIP either has IO or IO+BN finetuned. In addition to these two experimental settings, we also investigate fintuning input, output layers, and mean, variance of normalization layers, while fixing the convolution layer weights, normalization layer weights, and bias. The full experiment results with this extra setting is reported in Table 6 and 7. We can observe that compared with only finetuning input and output layers, updating mean and variance can also largely improve the performance of point-cloud recognition. As suggested in Section 5.2 in the main paper, updating mean and variance is to push the FIP models to generate point-cloud representation that is similar to image representation.

## A.3 ABLATION STUDY OF INFLATING TOWARDS DIFFERENT DIRECTIONS.

We conduct experiments of inflating filters along different directions, the illustration figure is shown in Figure 5, and the results are shown in Table 8. We find that the performance is different when using different inflation methods. In particular, with ResNet50 pretrained on ImageNet1K, inflating along x axis and the y axis leads to better performance compared with inflating along z axis for both FIP-IO and FIP-IO+BN. More importantly, the minimally finetuned FIP-IO+BN with inflating along the x and y axis even surpasses the training-from-scratch.

## A.4 ABLATION STUDY OF LOADING DIFFERENT STAGES OF IMAGE-PRETRAINED MODEL.

We investigate the effect of loading different subset of stages. The results are shown in Table 9. In detail, we load the pretrained weights partially while keeping the other weights randomly initialized.

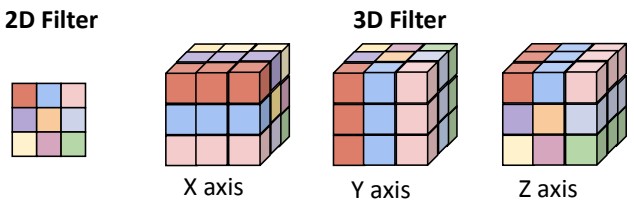

Figure 5: Visualization of filter inflation along different axis.

Table 8: ModelNet 3D Warehouse results (Top-1 accuracy) of partially finetuning ResNet50 pretrained on ImageNet1K with inflation along the x, y, z axis.

| Method | x axis | y axis | z axis |
|---|---|---|---|
| FIP-IO | 82.17 | 81.73 | 81.20 |
| FIP-IO+BN | 90.44 | 90.84 | 89.87 |
| From Scratch | | 90.32 | |

Table 9: ModelNet 3D Warehouse results (Top-1 accuracy) of finetuning ResNet50 pretrained on ImageNet1K with only subset of stages loaded.

| Loaded Stages | 1 | 1 2 | 1 2 3 | 1 2 3 4 | 2 3 4 | 3 4 | 4 |
|---|---|---|---|---|---|---|---|
| FIP-ALL | 89.91 | 90.36 | 90.64 | 90.92 | 91.09 | 90.19 | 89.99 |
| From Scratch | | | | 90.32 | | | |

We observe that excluding the weights of the first stage achieves the best performance, bringing 0.77 points improvement.

### A.5 AN INTUITION OF WHY THE IMAGE-PRETRAINED WEIGHTS CAN BE USED FOR POINT-CLOUD RECOGNITION.

In this subsection, we try to give an intuition of why the weights pretrained on images can be used for point-cloud tasks from the aspect of local feature projection. Let's only look at the one time convolution on a local point-cloud feature $X_{local} \in R^{C_{in} \times C_{out} \times K^3}$. In fact, a filter inflation can be represented as linear projection, given by $W_{2d} \otimes T = W_{3d}$, where $W_{2d} \in R^{C_{in} \times C_{out} \times K^2}$ is pretrained 2D weight, $W_{3d} \in R^{C_{in} \times C_{out} \times K^3}$ is the transformed 3D weight from $W_{2d}$, and $T \in R^{K^2 \times K^3}$ is the projection matrix. Therefore, for a local 3D feature $X_{local} \in R^{C_{in} \times C_{out} \times K^3}$, the convolution can be formed as $W_{3d} \otimes X_{local} = W_{2d} \otimes T \otimes X_{local} = W_{2d} \otimes X_{local-2d}$, where $X_{local-2d} \in R^{C_{in} \times C_{out} \times K^2}$ is the 2D feature projected from $X_{local}$. Essentially, applying the inflated filters on 3D point-clouds is equal to the local projection of 3D point-clouds to 2D then applying the pretrained 2D filters. A visual explanation is in Figure 6.

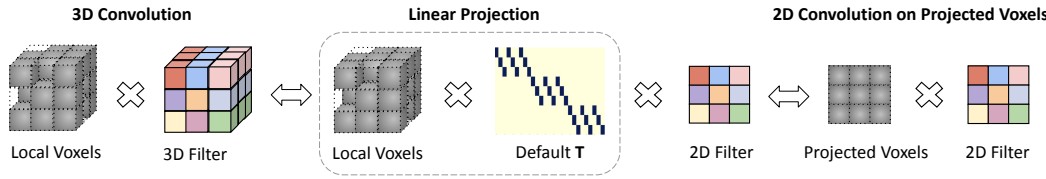

Figure 6: Visualization of filter inflation explained with linear projection matrix.

Table 10: ModelNet 3D Warehouse results (Top-1 accuracy) of finetuning ResNet18 pretrained on different Tiny-ImageNet.

| Method | Original image | Canny edge detection | Bilateral filters | Haar transformation |
|--------|----------------|----------------------|-------------------|---------------------|
| FIP-IO | 78.89 | 78.85 (**-0.04**) | 78.97 (**+0.08**) | 78.49 (**-0.40**) |

| Method | Shuffle patch 56 | Shuffle patch 28 | Shuffle patch 14 | Shuffle patch 7 |
|--------|------------------|------------------|------------------|-----------------|
| FIP-IO | 78.44(**-0.45**) | 78.00 (**-0.89**) | 76.21 (**-2.68**) | 74.60 (**-4.29**) |

### A.6 MORE VISUALIZATION OF NETWORK DISSECTION.

We present more visualization results based on the technique of network dissection, as shown in Figure 7. We can observe that for most cases, although there is no obvious visual concept transferred, the pretrained filters are prone to cluster similar objects on ModelNet 3D Warehouse dataset.

### A.7 USING DIFFERENT IMAGE PROCESSING METHODS FOR PRETRAINING.

To further investigate the image-to-point-cloud transfer in a more rigorous manner, we design a new set of experiments and report them in Appendix A.7 in the paper revision. The basic idea is that we perform a transformation (such as a low-pass filter) on the original image to destroy certain properties (high frequency components). Then, we train a ResNet18 on the transformed images, then transfer to point-cloud recognition. This way, we can better attribute which parts of the images are useful when transferred to point-cloud recognition. We perform the following transformations. Implementation code is taken from external sources[8][9][10].

1) Canny edge detection (Canny, 1986) transforms the images into high-frequency edge maps, discarding the low-frequency such as the surface of objects;

2) Bilateral filters (Tomasi & Manduchi, 1998) preserve the low-frequency properties of images while filtering out the high-frequency noise. Note that the high-frequency edges are still preserved and even sharpened;

3) Haar wavelet image compression (Talukder & Harada, 2010). The wavelet transform divides the information of an image into approximation and detailed sub-signals. If the detail is small, then it would be thrown;

4) Patch shuffling divides an image into patches with different patch sizes, shuffle the patches, and rearrange them into a new image. This transformation preserves the statistics of the images, while destroying the original order of the pixels. "Shuffle patch 56" means the patch size is $56 \times 56$.

The results are reported in the Table 10. Specifically, the edge images from canny edge detection are dramatically different from original images while keeping minimal performance drop. The bilateral filters discard the high-frequency noise, sharpening the high-frequency edge lead to the best performance, even outperforming the result of finetuning from the original image pretraining. As details of images are thrown (haar transformation), the performance slightly drops. On the other hand, shuffling the patches results in worse performance as the patch size gradually becomes small, which indicates that the order of the pixels are important to the transfer.

Therefore, we may infer that the structured (ordered) edges are important to the transfer.

### A.8 DETAILS OF FIRST-WASSERSTEIN DISTANCE ON RESNET18.

We list all the results in each layer of ResNet18, as shown in Table 11. We can observe that for each layer, the first-wasserstein distance is largely reduced when finetuning more parameters.

---

[8]https://www.geeksforgeeks.org/implement-canny-edge-detector-in-python-using-opencv/
[9]https://docs.opencv.org/3.4/d4/d13/tutorial_py_filtering.html
[10]https://medium.com/@digitalpadm/image-compression-haar-wavelet-transform-5d7be3408aa

Table 11: First-wasserstein distance between image model features and different 3D model features of all 16 layers in ResNet18

| Image Model | FIP/IO | FIP/IO-BN | FIP/ALL | Training-from-scratch |
|---|---|---|---|---|
| Average | $2.1 \times 10^2$ | 0.27 | 0.093 | 0.15 |
| Layer 1 | 1.9 | 0.2 | 0.094 | 0.12 |
| Layer 2 | 1.1 | 0.064 | 0.11 | 0.048 |
| Layer 3 | 1.3 | 0.24 | 0.068 | 0.095 |
| Layer 4 | 2.6 | 0.14 | 0.051 | 0.077 |
| Layer 5 | 1.1 | 0.23 | 0.039 | 0.082 |
| Layer 6 | 3 | 0.21 | 0.025 | 0.063 |
| Layer 7 | 5.4 | 0.29 | 0.05 | 0.15 |
| Layer 8 | 13 | 0.26 | 0.051 | 0.097 |
| Layer 9 | 12 | 0.28 | 0.04 | 0.095 |
| Layer 10 | 26 | 0.2 | 0.077 | 0.13 |
| Layer 11 | 54 | 0.27 | 0.032 | 0.13 |
| Layer 12 | $2.2 \times 10^2$ | 0.25 | 0.076 | 0.094 |
| Layer 13 | $1.9 \times 10^2$ | 0.29 | 0.028 | 0.11 |
| Layer 14 | $7.7 \times 10^2$ | 0.16 | 0.075 | 0.041 |
| Layer 15 | $9.5 \times 10^2$ | 0.29 | 0.081 | 0.088 |
| Layer 16 | $1.2 \times 10^3$ | 0.89 | 0.59 | 1 |

## A.9  DETAILS OF USED ARCHITECTURES.

```
1  Class 3DRes_cls(nn.Module):
2      def __init__(self, res_block):
3          super().__init__() # res_block means the residual block as same
   as the conventional ResNet.
4          self.input_layer = nn.Sequential(
5              sparse_conv3d(input_dim, layer1_Idim, k=3, s=1),
6              sparse_bn(layer1_Idim))
7
8          self.layer1 = inflated_resnet_layer1(res_block, layer1_Idim,
   layer1_Odim)
9          self.layer2 = inflated_resnet_layer2(res_block, layer2_Idim,
   layer2_Odim)
10         self.layer3 = inflated_resnet_layer3(res_block, layer3_Idim,
   layer3_Odim)
11         self.layer4 = inflated_resnet_layer4(res_block, layer4_Idim,
   layer4_Odim)
12
13         self.output_layer = nn.Sequential(
14             global_average_pooling,
15             nn.Linear(layer4_Odim, class_num),
16             nn.bn(class_num))
17
18     def forward(self, x):
19         x = self.input_layer(x)
20         x = self.layer1(x)
21         x = self.layer2(x)
22         x = self.layer3(x)
23         x = self.layer4(x)
```

```
24            return self.output_layer(x)
```

Listing 1: Pseudo code of inflated ResNet with linear input and output for classification

```
1  Class 3DRes_seg(nn.Module):
2      def __init__(self, res_block):
3          super().__init__() # res_block means the residual block as same
   as the conventional ResNet.
4          self.input_layer = nn.Sequential(
5              sparse_conv3d(input_dim, layer1_Idim, k=3, s=1),
6              sparse_bn(layer1_Idim),
7              sparse_ReLU(True),
8              sparse_conv3d(layer1_Idim, layer1_Idim, k=3, s=1),
9              sparse_bn(layer1_Idim),
10             sparse_ReLU(True),
11             sparse_conv3d(layer1_Idim, layer1_Idim, k=3, s=2),
12             sparse_bn(layer1_Idim),
13             sparse_ReLU(True))
14
15         self.layer1 = inflated_resnet_layer1(res_block, layer1_Idim,
   layer1_Odim)
16         self.layer2 = inflated_resnet_layer2(res_block, layer2_Idim,
   layer2_Odim)
17         self.layer3 = inflated_resnet_layer3(res_block, layer3_Idim,
   layer3_Odim)
18         self.layer4 = inflated_resnet_layer4(res_block, layer4_Idim,
   layer4_Odim)
19
20         self.up1 = sparse_deconv(layer4_Odim, layer4_Odim, k=2, s=2),
21         self.decode1 = self.Sequential(
22             res_block(layer4_Odim+layer3_Odim, layer3_Odim),
23             res_block(layer3_Odim, layer3_Odim))
24
25         self.up2 = sparse_deconv(layer3_Odim, layer3_Odim, k=2, s=2)
26         self.decode2 = self.Sequential(
27             res_block(layer3_Odim+layer2_Odim, layer2_Odim),
28             res_block(layer2_Odim, layer2_Odim))
29
30         self.up3 = sparse_deconv(layer2_Odim, layer2_Odim, k=2, s=2)
31         self.decode3 = self.Sequential(
32             res_block(layer2_Odim+layer1_Odim, layer1_Odim),
33             res_block(layer1_Odim, layer1_Odim))
34
35         self.up4 = sparse_deconv(layer1_Odim, layer1_Odim, k=2, s=2)
36         self.decode4 = self.Sequential(
37             res_block(layer1_Odim+layer1_Odim, layer1_Odim),
38             res_block(layer1_Odim, layer1_Odim))
39
40         self.output_layer = nn.Sequential(
41             nn.Linear(layer1_Odim, class_num))
42
43     def forward(self, x):
44         x_i = self.input_layer(x)
45         x1 = self.layer1(x_i)
46         x2 = self.layer2(x1)
47         x3 = self.layer3(x2)
48         x4 = self.layer4(x3)
49
50         x3_ = self.decoder1(cat(x3, self.up1(x4)))
51         x2_ = self.decoder2(cat(x2, self.up2(x3_)))
52         x1_ = self.decoder3(cat(x1, self.up3(x2_)))
53         xi_ = self.decoder4(cat(x_i, self.up4(x1_)))
54         return self.output_layer(xi_)
```

Listing 2: Pseudo code of inflated ResNet for segmentation

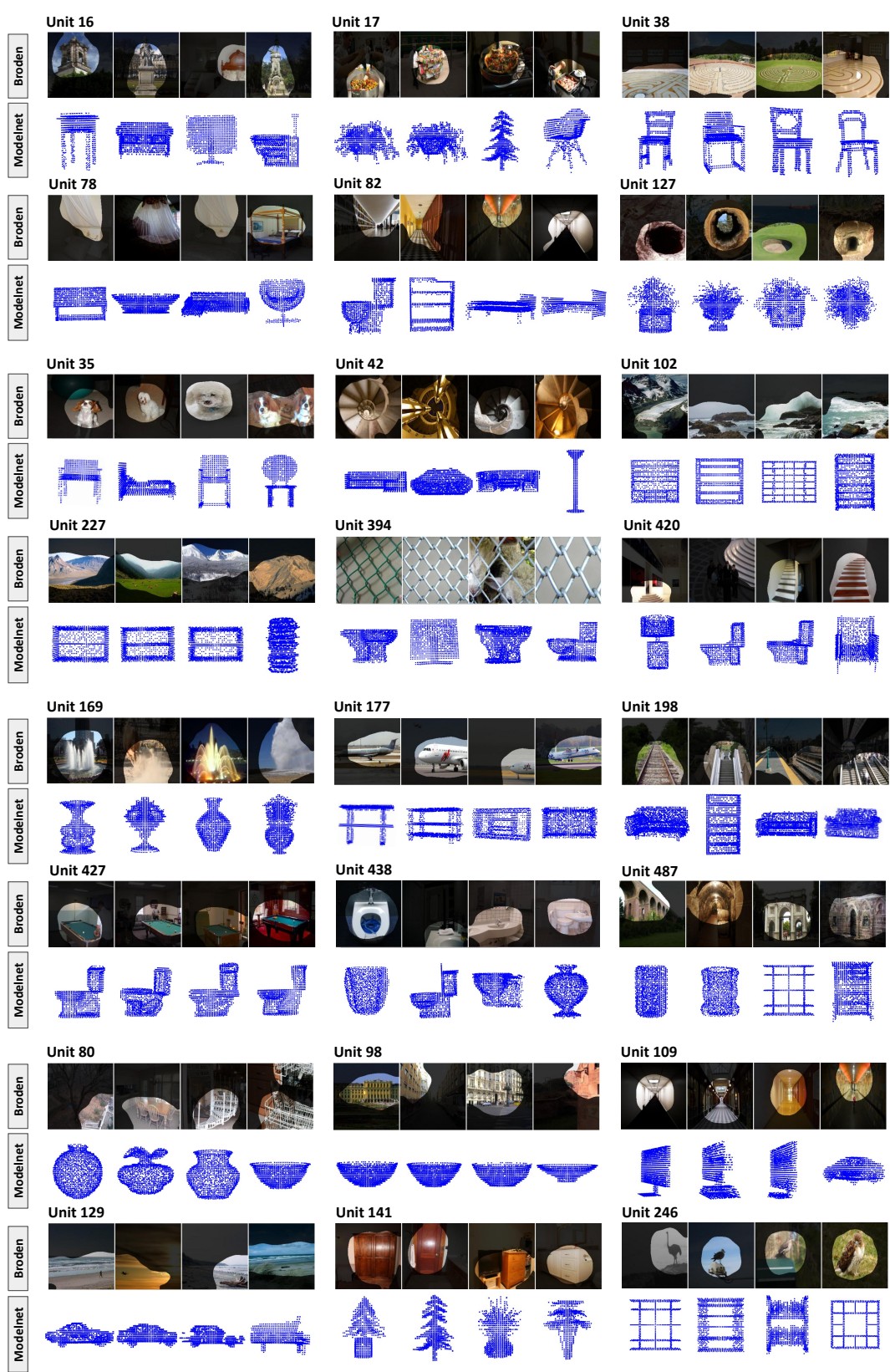

Figure 7: More network dissection results.

