# OpenReview forum: "Image2Point: 3D Point-Cloud Understanding with 2D Image Pretrained Models"
_ICLR.cc/2022/Conference — ICLR 2022 Submitted_

### Official Review · Reviewer_SBBb · 2021-11-01

**Correctness:** 4
**Technical Novelty And Significance:** 4
**Empirical Novelty And Significance:** 4
**Recommendation:** 6
**Confidence:** 5

**Main Review:**

Strength:

The paper is simple yet very interesting, which simple inflate the 2D convolution kernels to 3D kernels to learn the representation of the 3D point clouds. 2D images and 3D point cloud are of great differences. Thus, it seems to impractical to simply performed the inflation and make it work. However, the authors proposed one simple idea and demonstrate that the idea does work.  I like the simple idea and would like to recommend the acceptance.


Weakness:
1. Some detailed information is missing. Therefore, I am not able to follow the detailed process. In Table 1, the authors list the performances of pointnet++ and the performances. I am curious how to performance the inflation in point net++
2. The authors perform the experiments on the classification and segmeantion. What about the performances on detection task.




=========
Post-rebuttal

I have read the authors rebuttal as well as the other fellow reviewers' comments and the corresponding discussions.

I agree with the other fellow reviewers' comment, 2D inflating to 3D has been studied in video action recognition. However, to me, inflating 2D CNN of image for 3D pointcloud is new. Therefore, I do think the paper is of novelty.
Also I agree with the other reviewers' comments that the discussion of why the proposed method works does not been clearly addressed.


Considering this, I am lowering my rating from 8 to 6.


**Summary Of The Paper:**

The paper proposed to perform the 3D point cloud understanding based on the pretrained 2D image models. Simple inflation technical, same as the work in I3D, are used to convert the 2D convnet to 3D convnet.

Extensive experiments performed on the point cloud classification and segmentation tasks demonstrate the effectiveness of the work.


**Summary Of The Review:**

It seems to the reviewer that this is the first paper to use 2D image pertained models for 3D point cloud understanding. Although the idea is simple, the work is interesting and inspiring. Therefore, I am recommending the acceptance.

---

> ### Author Response · Authors · 2021-11-20
> **Response**
>
> We sincerely appreciate your highly acknowledgement of our work! The image representation transfer is able to generalize to different backbones, including ResNet, PointNet++, Vision Transformer, and SimpleView.
>
> Regarding the PointNet++, it doesn’t need any inflation. PointNet++ mimics the convolution by sampling the central points, grouping nearby points, and aggregating the local features. Therefore, it is natural to apply it on the images for image pretraining. We break the images into pixels, and regard each pixel as a point, then train the PointNet++ on the ImageNet1K dataset. The details are illustrated in our Appendix A.1. After pretraining, we directly finetune on point-cloud datasets without changing the model.
>
>
> Regarding the detection result, thanks for your suggestion of doing experiments on detection. Modern object detection models are not very different from segmentation models, in the sense that an object detection task can be formulated as a point-wise/pixel-wise prediction/regression task [1]. Due to the time and computational limitations, it is hard to conduct experiments at this stage, we would report the results as soon as we get a chance to conduct the experiments.
>
> [1] CenterNet: Objects as Points. Zhou et al. CVPR 2019.

---

> ### Author Response · Authors · 2021-11-30
> **Response about why it works**
>
> Sincerely thanks for the further comment.
>
> We acknowledge that we did not fully address them since such an explanation still requires more exploration. However, we would emphasize that we are the first to apply the analysis method of the shape-texture analysis, net dissection, and feature distribution in image-to-point-cloud transfer, and we did more exploration of why it works during the rebuttal period. It includes
> * Conducting experiments based on explainable operations like haar transformation, canny edge detection, bilateral filter, and patch suffling, by which we conclude that the structured edges are transferred. The reviewers may refer to our appendix A.7.
> * We provide an intuitive that inflating the 2D weight is essentially equal to locally projecting the 3D feature into 2D. Connecting to our analysis of BN, after tuning the BN, the pretrained 2D weights potentially apply to the 3D features with similar element-wise distribution to 2D. The reviewers may refer to A.5.

---

### Official Review · Reviewer_2iHv · 2021-11-02

**Correctness:** 3
**Technical Novelty And Significance:** 1
**Empirical Novelty And Significance:** 3
**Recommendation:** 6
**Confidence:** 3

**Main Review:**

Pros: The idea of transfer learning between image and point cloud data seems to be novel and surprising.

Cons:
1. The idea of inflating model weight from 2D to 3D is not new[1].
2. The performance on point cloud segmentation on SemanticKitti dataset, while is reasonably well, still has gap to the state-of-the-art(This one is minor)
3. The effectiveness of pretrained weight would be more convincing if the author could do further ablation study to initialize new model with expanding subsets of layers and exploring the effect on model performance. It will indicate which part of the pertained model is really useful in this transferred learning setting.

=====
Update after rebuttal: After reading the author's response and the new experimental results amended by the author, I raise my score to 6.

[1] Carreira, Joao, and Andrew Zisserman. "Quo vadis, action recognition? a new model and the kinetics dataset." proceedings of the IEEE Conference on Computer Vision and Pattern Recognition. 2017.

**Summary Of The Paper:**

This paper described an experiment of transfer learning between image and point cloud data. Despite of apparent dissimilarity between the two domain, the author was able to use structure and weights from pertained models in image domain to get reasonable performance in point cloud tasks like classification and segmentation. The author also made comparison of using pertained weights versus training from scratch, the result shows training pertained weights improves data efficiency. The proposed method is also proven to be useful when the training data is limited by training under few-shot learning setting.


**Summary Of The Review:**

Overall the paper has interesting experimental result on transfer learning between image and point cloud data. The authors did a good job explaining the idea, concepts, procedures and experiments, showing the effectiveness of the transferred model on performance. data efficiency and usefulness on few-shot setting. The experiment itself could be further extended though.

---

> ### Author Response · Authors · 2021-11-20
> **Response**
>
> We sincerely thank you for your comments. Your concerns are responded as the following.
>
> Q1: The idea of inflating model weight from 2D to 3D is not new.
>
> A1:
> We agree that the inflation technique used in the paper is not new, since it has been widely used in video understanding [1] and medical image analysis [2], as discussed in our related work (Section 2.3). However, we think the highlight of the paper is to share a surprising discovery. Given the huge gap between images (dense, regular, and RGB pixels) and point-clouds (sparse, unordered, and x, y, z points lying on object surface), it seems quite unlikely that models trained on images could transfer to point-cloud. Nevertheless, our experiments show not only such transfer works, but it can help to boost the point-cloud models’ performance effectively.
>
> Q2: Gap between the SOTA on SemanticKITTI
>
> A2:
> We try to use the simplest method to explore the very surprising discovery. Sprinting to the SOTA, especially on such a challenging task, is our future work. Some hints showing that this direction leading to SOTA is very promising: the SOTA method SimpleView[1] on the ModelNet 3D Warehouse can be further improved by 0.5 points with ImageNet1K pretraining; the mainstream methods on SemanticKITTI, projection methods (HRNet), can be improved by 3.4 points and 5.25 points with pretraining on ImageNet1K and CityScapes, respectively. We also find that as the pretrained dataset gets larger, FIP achieves larger improvements, e.g., using ImageNet21K achieves the best on top of ResNet50. Moreover, as the pretrained dataset domain becomes similar to the point-cloud dataset domain, the performance can be improved more significantly, e.g., CityScapes pretraining brings 5.25 points improvement on the SemanticKITTI point-cloud dataset, while ImageNet1K brings 3.41 points improvement.
>
> Q3: “Do further ablation study to initialize new model with expanding subsets of layers and exploring the effect on model performance?”
>
> A3:
> Thank you for your insightful suggestions. We analyze it as shown in the table below. We gradually load and unload the pretrained weights stage by stage. As loading pretrained weights of all the stages excluding the first stage, the performance achieves the best.
>
> | ResNet50 ImageNet1K  | stage 1 | stage 1 2 | stage 1 2 3 | stage 1 2 3 4 | stage 2 3 4 | stage 3 4 | stage 4 |
> | ----- | ----- | ----- | ----- | ----- | ----- | ----- | ----- |
> | FIP-ALL | 89.91 | 90.36 | 90.64 | 90.92 | 91.09 | 90.19 | 89.99 |
> | From scratch |||| 90.32 |
> Table.1. ModelNet 3D Warehouse results (Top-1 accuracy) of finetuning ResNet50 pretrained on ImageNet1K with only subset of stages loaded.
>
> [1] Revisiting point cloud shape classification with a simple and effective baseline. Ankit Goyal, Hei Law, Bowei Liu, Alejandro Newell, and Jia Deng. ICML2021.

---

> > ### Comment · Reviewer_2iHv · 2021-11-30
> > **Post-rebuttal**
> >
> > Thanks the author for addressing the concerns. The ablation study result in the rebuttal looks particularly intriguing as it may suggest the weights of higher level layers, while has larger receptive field, has more impact in the transfer learning. This result should definitely be added to the paper. Actually I was hoping this kind of ablation study to be applied to experiments on all datasets in this paper. Considering the novelty of the experiment and interesting results shown, I am willing to update the score from 5 to 6.

---

### Official Review · Reviewer_KdMv · 2021-11-02

**Correctness:** 4
**Technical Novelty And Significance:** 2
**Empirical Novelty And Significance:** 2
**Recommendation:** 6
**Confidence:** 4

**Main Review:**

Strengths
- The paper proposes a model to connect 2D and 3D representations in transfer learning.
- Detailed experiments show that the proposed model improves the performance and data efficiency compared to training from scratch.

Weaknesses
- Will larger dataset leads to better performance? A lot of comparisons between results from datasets of different size are missing in Table 1.
- The paper shows that using 2D pretrained features improve performance. However, results of state-of-the-art task specific methods should also be listed. In addition, it would be better to compare with different self-supervised learning method.
For example, for shape classification on ModelNet40:
Foldingnet: Point cloud auto-encoder via deep grid deformation. CVPR 2018
Pointcontrast: Unsupervised pretraining for 3d point cloud understanding. ECCV 2020
-  Explanation why this kind of inflating 2D filter to 3D is reasonable. There is a huge domain gap between 2D dataset and 3D dataset. It's still unclear why this kind of transfer is reasonable. It would be better to add theoretical analysis besides experiments.

**Summary Of The Paper:**

The paper proposes to transfer the 2D image features for 3D point cloud understanding tasks. Specifically, the proposed method first inflates the 2D convolution filters from a pretrained model to 3D and then optimizes only input, output and optionally batch normalization layer.  Detailed experiments show that the proposed finetuned-image-pretrained models can improve the performance and data efficiency.


**Summary Of The Review:**

Although the experiments in the paper show that the proposed model can improve the performance and data efficiency compared to training from scratch, the overall experiments results are not strong enough. In addition, it's still unclear why this kind of 2D to 3D transfer is reasonable. I am inclined to reject.

Update: After reading the feedbacks from the author, I raise my score to 6.

---

> ### Author Response · Authors · 2021-11-20
> **Response**
>
> We sincerely appreciate your comment. Your concerns are addressed below.
>
> Q1: Will larger datasets lead to better performance?
>
> A1:
> Yes. This is mainly shown in the Table below. Specifically, as the dataset size increases from 0.1M (Tiny-ImageNet) to 1.2M (ImageNet1K), FIP-ALL performs 0.28 points better. When the dataset size increases from 1.2M (ImageNet1K) to 14M (ImageNet21K), the performance gets further improved. We only conduct this experiment based on ResNet50, since 1) it is representative enough; 2) we do not have pretrained weights for other models and do not have sufficient compute resources to pretrain all these models on three image datasets.
>
> To better show the relationship between dataset size and performance, in Figure 2 (b) in our revision (Figure 2 (a) in our initial submission) we show the performance of using different pretrained datasets with different sizes under the setting of partially finetuning (FIP-IO and FIP-IO+BN).
>
> | ResNet50 | TinyImageNet | ImageNet1K | ImageNet21K |
> | ----- | ----- | ----- | ----- |
> | FIP-ALL | 90.64 | 90.92 | 91.05 |
> | From scratch || 90.32 |
> Table.1. ModelNet 3D Warehouse results (Top 1 accuracy) of finetuning ResNet50 pretrained on different sizes of ImageNet.
>
> Q2: Comparison with SOTA, and other self-supervised methods
>
> A2:
> We list the SOTA in table 1 in the revision. In particular, on top of the well-known SimpleView [1], the image pretraining boosts 0.5 points (achieving 93.8% top 1 accuracy on ModelNet 3D Warehouse dataset), outperforming a wide range of 2021 SOTAs e.g. RSMix [2] (93.5 top-1 accuracy), Point Transformer [3] (93.7 top-1 accuracy), DRNet [4] (93.1 top-1 accuracy), and PointCutMix [5] (93.4 top-1 accuracy).
>
> We also thank you for pointing out that we should compare with self-supervised learning methods like PointContrast. The results are listed below. The ImageNet1K pretraining leads to at least 0.64 points improvement than PointContrast, indicating that our method is indeed effective. The detailed experiment setting is illustrated below too.
>
> | Method | Top-1 Accuracy |
> | ----- | ----- |
> | PointContrast (Hardest contrastive) | 90.15 (+0.20) |
> | PointContrast (NCE) | 90.24 (+0.29) |
> | Imagenet1K pretraining (ours)| 90.88 (+0.93) |
> | Training-from-scratch | 89.95 |
> Table.2. ModelNet 3D Warehouse results (Top-1 accuracy) of finetuning ResNet34 pretrained on ImageNet1K or PointContrast.
>
> Since PointContrast didn’t conduct experiments on the ModelNet 3D Warehouse dataset, we directly use their used backbone 3D ResNet34 and train the model with the same training recipe. We also directly take their model pretrained on the ScanNet using PointInfoNCE loss. The code and the pretrained model are from https://github.com/facebookresearch/PointContrast. For our image-pretrained model, we use the same backbone as PointContrast but compress it into 2D, then we train it on ImageNet1K with the standard training recipe https://github.com/pytorch/examples/blob/master/imagenet/main.py. Finally, we inflate the image-pretrained ResNet34 into 3D, and then finetune it on ModelNet 3D Warehouse.
>
> Q3: It would be better to add theoretical analysis besides experiments.
>
> A3:
> Thanks for your suggestion. We try to provide an intuition as the following. Let's only look at the one time convolution on a local point-cloud feature $X_{local} \in R^{C_{in} \times C_{out} \times K^3}$. In fact, a filter inflation can be represented as linear projection, given by $W_{2d} \otimes T = W_{3d}$ , where $W_{2d} \in R^{C_{in} \times {C_{out}} \times {K^2}}$ is pretrained 2D weight, $W_{3d} \in R^{C_{in} \times C_{out} \times K^3}$ is the transformed 3D weight from $W_{2d}$, and $T \in R^{K^2 \times K^3}$ is the projection matrix. Therefore, for a local 3D feature $X_{local} \in R^{C_{in} \times C_{out} \times K^3}$, the convolution can be formed as $W_{3d} \otimes X_{local} = W_{2d} \otimes T \otimes X_{local} = W_{2d} \otimes X_{local-2d}$, where $X_{local-2d} \in R^{C_{in} \times C_{out} \times K^2}$ is the 2D feature projected from $X_{local}$. Essentially, applying the inflated filters on 3D point-clouds is equal to the local projection of 3D point-clouds to 2D then applying the pretrained 2D filters.
>
> [1] Revisiting point cloud shape classification with a simple and effective baseline. Ankit Goyal, Hei Law, Bowei Liu, Alejandro Newell, and Jia Deng. ICML2021.
>
> [2] Regularization Strategy for Point Cloud via Rigidly Mixed Sample. Dogyoon Lee, Jaeha Lee, Junhyeop Lee, Hyeongmin Lee, Minhyeok Lee, Sungmin Woo, Sangyoun Lee. CVPR 2021.
>
> [3] Point Transformer. Hengshuang Zhao, Li Jiang, Jiaya Jia, Philip Torr, Vladlen Koltun. ICCV 2021.
>
> [4] Dense-Resolution Network for Point Cloud Classification and Segmentation. Shi Qiu, Saeed Anwar, Nick Barnes. WACV 2021.
>
> [5] PointCutMix: Regularization Strategy for Point Cloud Classification. Jinlai Zhang, Lyujie Chen, Bo Ouyang, Binbin Liu, Jihong Zhu, Yujing Chen, Yanmei Meng, Danfeng Wu. arXiv preprint 2021.

---

### Official Review · Reviewer_DFK1 · 2021-11-03

**Correctness:** 3
**Technical Novelty And Significance:** 3
**Empirical Novelty And Significance:** 4
**Recommendation:** 6
**Confidence:** 5

**Main Review:**

Strength:

- This paper investigates a new problem of transferring 2D pre-training to 3D.
- The presentation is clear and easy to follow.
- The experiments are extensive. The performance gain using the 2D pre-training is notable and non-trivial, showing the proposed weight transferring technique has the ability to convert useful learned information on 2D to 3D.

Weakness:

- It's good that the authors showed some visualizations to help understand the proposed framework. However, as the authors pointed out, the visualization cannot show why the transfer works or what information can be transferred.
- The last sentence on page 8 is questionable: the conclusion of "shape representations are better transferred from image to point-cloud" cannot be inferred from overall dataset performances. Also, what does "shape representations" mean?


**Summary Of The Paper:**

This paper proposed a pipeline for transferring convolutional network weights that are pre-trained on 2D images to 3D convolution networks. The proposed approach is to inflate the 2D kernels to 3D kernels similar to video models. The experiments are conducted on 2D image datasets such as ImageNet, and 3D datasets such as ModelNet.

**Summary Of The Review:**

This paper proposed a solution to a new problem of transferring 2D weights to 3D. The solution works well on many dataset pairs. Though some claims made in the paper are questionable, this paper deserves to be published at the conference.

---

> ### Author Response · Authors · 2021-11-20
> **Response**
>
> We sincerely thank you for your acknowledgment of our idea, our presentation, and our experiments. We also appreciate you for pointing out the issues about why it works.
>
> Regarding shape representation, as far as we know, there is not a formal and rigorous definition to divide shape vs. texture in the computer vision community. Our notion of shape vs. texture mainly comes from [1]. They propose the model training on the original ImageNet dataset is more biased to the texture, thus designing a Stylized ImageNet dataset to force the model biased to the shape. However, they didn’t explicitly and mathematically define the shape representation.
>
> Regarding the questionable conclusion, sorry for the confusion. Based on the shape-texture division of image representation from [1], we leverage the “shape-biased” pretrained model provided by [1] and the “texture-biased” pretrained model provided by [2]. As shown in Table 4 of our submission, the shape-based models work better than texture-bias pretrained models on point-cloud recognition, thus a more precise conclusion is that the shape representation is transferred and benefits point-cloud tasks.
>
> To make clear what is transferred mathematically, we leverage mathematically explainable image processing operations, including canny edge detection, bilateral filters, haar transformation, and patch shuffling. The results are reported in the Appendix A.7 of our revision.
>
> [1] ImageNet-trained CNNs are biased towards texture; increasing shape bias improves accuracy and robustness. Robert Geirhos, Patricia Rubisch, Claudio Michaelis, Matthias Bethge, Felix A. Wichmann, Wieland Brendel. ICLR 2019.
>
> [2] Approximating CNNs with Bag-of-local-Features models works surprisingly well on ImageNet. Wieland Brendel, Matthias Bethge. ICLR2019.

---

### Official Review · Reviewer_aajt · 2021-11-08

**Correctness:** 3
**Technical Novelty And Significance:** 2
**Empirical Novelty And Significance:** 2
**Recommendation:** 6
**Confidence:** 3

**Main Review:**


Strengths:
1. Idea is good and helpful for the 3D point cloud field.
2. The experiments are designed to test different advantage of this idea. The results support the arguments.

Weaknesses / questions:
1. Since such idea has already used in other 2D-3D domain, the novelty is limited. Such paper needs extra emphasis on discussing why it works and ablation studies.
  - In discussion section, the author provides some experiments, but the result seems quite random (figure 4). The "shape representation" are better transferred seems make sense, but why? The author tried to explain the finding but seems not quite convincing.
  - When designing inflation, is there a difference on which axis to inflate. x, y or z axis. This is different to video since 3 axis is symmetric.
2. The biggest concern is the parameter sizes and network structure.
  - In 4.1, the author did not compare the parameter size. I doubt the fairness of the comparison with the baseline since it is possible that the performance gain is fully from the increasing parameters.
  - In 4.2, there is a possibility that the ResNet structure is not good structure to train on point cloud making it quite bad on scratch training. For pointnet++, the training method is quite strange.
3. The writing for this paper is interesting... not formal enough. The author used 10 "surprising(ly)" in this paper...

**Summary Of The Paper:**

This paper introduced "inflation" (2D CNN kernel to 3D one by repeating on the 3rd axis) to transform a 2D image pre-train backbone into a 3D version so that 3D point cloud task can be benefit from 2D image pretraining. Detailed experiments are performed to address this idea. Results show minimal finetuning efforts can achieve competitive performance on 3D tasks. The authors are kind of "surprised" by these results.

**Summary Of The Review:**

This is an interesting idea. This paper provided detailed experiments to test different advantage of this idea. However, the discussion on why it work seems not quite convincing. Also, part of the experiments missed some details. Some conclusions may not correct.
We expect the author provided more convincing discussion and more comprehensive comparison detail.

---

> ### Author Response · Authors · 2021-11-20
> **Response to novelty and shape transfer**
>
> We sincerely appreciate your acknowledgment of our idea and the helpfulness to the 3D point-cloud field. We respond to your concerns as below.
>
> Q1: “Such idea is already used in other 2D-3D domain, the novelty is limited”
>
> A1:
> We agree that the inflation technique used in the paper is not novel, since it has been widely used in video understanding [1] and medical image analysis [2], as discussed in our related work (Section 2.3). However, we think the novelty of the paper is to share a surprising discovery. Given the huge gap between images (dense, regular, and RGB pixels) and point-clouds (sparse, unordered, and x, y, z points lying on object surface), it seems quite unlikely that models trained on images could transfer to point-cloud. Nevertheless, our experiments show not only such transfer works, but it can help to boost the point-cloud models’ performance effectively. We would like to gently point out that the novelty of this discovery is also recognized by other reviewers (DFK1, 2iHV, SBBb).
>
> Q2: Need extra emphasis on discussing why it works and ablation studies. The conclusion about shape transferring is not convincing.
>
> A2:
> We agree with you that we should provide deeper analyses on why it works. In the submission, we provide three methods, net dissection, shape-texture analysis, and feature distribution analysis, all of which we believe to be novel methods to analyze the transfer between images and point-clouds.
>
> Regarding the shape-texture analysis, in fact, as far as we know, there is no rigorous definition for shape vs. texture. In Section 5.1 of our submission, we follow a previous work [5] that divides the image representation as shape representation and texture representation, and use their provided pretrained models to explore what is transferred.
>
> To further investigate the image-to-point-cloud transfer in a more rigorous manner, we design a new set of experiments and report them in Appendix A.7 in the paper revision. The basic idea is that we perform a transformation (such as a low-pass filter) on the original image to destroy certain properties (high-frequency components). Then, we train a ResNet18 on the transformed images, then transfer to point-cloud recognition. This way, we can better attribute which parts of the images are useful when transferred to point-cloud recognition. We perform the following transformations:
>
> * Canny edge detection [6] transforms the images into high-frequency edge maps, discarding the low-frequency such as the surface of objects.
> * Bilateral filters [7] preserve the low-frequency properties of images while filtering out the high-frequency noise. Note that the high-frequency edges are still preserved and even sharpened;
> * Haar wavelet image compression [8]. Wavelet transform divides the information of an image into approximation and detailed sub-signals. If the detail is small, then it would be thrown.
> * Patch shuffling divides an image into patches with different patch sizes, shuffles the patches, and rearranges them into a new image. This transformation preserves the statistics of the images, while destroying the original order of the pixels. “Shuffle patch 56” means the patch size is 56 x 56.
>
> The results are reported in the table below. Specifically, the edge images from canny edge detection are dramatically different from original images while keeping minimal performance drop. The bilateral filters discard the high-frequency noise, sharpening the high-frequency edge leads to the best performance, even outperforming the result of finetuning from the original image pretraining. As details of images are thrown (haar transformation), the performance slightly drops. On the other hand, shuffling the patches results in worse performance as the patch size gradually becomes small, which indicates that the order of the pixels is important to the transfer.
>
> Therefore, we conclude that the structured (ordered) edges are important to the transfer.
>
> | Method | Original image | Canny edge detection | Bilateral filters | Haar transformation | Shuffle patch 56 | Shuffle patch 28 | Shuffle patch 14 | Shuffle patch 7 |
> | ----- | ----- | ----- | ----- | ----- | ----- | ----- | ----- | -----
> | FIP-IO | 78.89 | 78.85 (-0.04) | 78.97 (+0.08) | 78.49 (-0.40) | 78.44 (-0.45) | 78.00 (-0.89) | 76.21 (-2.68) | 74.60 (-4.29)
>
> Table.1. ModelNet 3D Warehouse results (Top-1 accuracy) of finetuning ResNet18 pretrained on different Tiny-ImageNet.
>
> The interpretation of deep learning models, or especially here, the interpretation of the cross-modal deep learning models, requires long-term exploratory by the whole research community. We hope these attempts can inspire the research community to explore deeper in this direction.

---

> ### Author Response · Authors · 2021-11-20
> **Response to inflation to different directions, parameters, and PointNet++**
>
> Q3: x, y, z inflation.
>
> A3: Thanks for your insightful suggestion. We conduct experiments of inflating filters along with different directions, and the results are shown below. We find that the performance is indeed different when using different inflation directions. In particular, with ResNet50 pretrained on ImageNet1K, inflating along x axis and the y axis leads to better performance compared with inflating along z axis for both FIP-IO and FIP-IO+BN. More importantly, the minimally finetuned FIP-IO+BN with inflating along the x and y axis even surpasses the training-from-scratch.
>
> | ResNet50 ImageNet1K | x axis | y axis | z axis |
> | ----- | ----- | ----- | ----- |
> | FIP-IO | 82.17 | 81.73 | 81.20 |
> | FIP-IO+BN | 90.44 | 90.84 | 89.87 |
> | From scratch || 90.32 |
> Table.2. ModelNet 3D Warehouse results (Top-1 accuracy) of partially finetuning ResNet50 pretrained on ImageNet1K with inflation along the x, y, z axis.
>
> Q4: Performance gain is caused by more parameters.
>
> A4:
> Parameters are not usually positively related to the performance of point-cloud understanding models. For example, inflated ResNet50 contains 44.35x more parameters than PointNet++, yet the performance is still lower than PointNet++. Besides, ResNet152 and ResNet101x2 with training-from-scratch also perform worse than the lighter ResNet50. Even for the most classical methods, PointNet is 2.59 times larger than PointNet++, but the performance is worse. We also try to increase the parameters of PointNet by 9 times, also find the performance is not obviously improved. Therefore, the increasing of parameters doesn’t necessarily bring performance gain.
>
> More importantly, we compare Trainable parameters w.r.t Performance, as shown below:
>
> | Acc / Trainable parameters | ResNet18 (ImageNet1K pretrain) | ResNet50 (ImageNet21K pretrain) | ResNet152 (ImageNet1K pretrain) |
> | ----- | ----- | ----- | ----- |
> | FIP-IO | 71.03 / 0.026M (0.07%) | 73.74 / 0.088M (0.1%) | 64.63 / 0.088M (0.06%) |
> | FIP-IO+BN | 88.57 / 0.040M (0.1%) | 90.8 / 0.19M (0.3%) | 90.44 / 0.39M (0.3%) |
> | FIP-ALL | 90.52 / 34.4M (100%) | 91.05 / 65.65M (100%) | 91.09 / 136.96M (100%) |
> | Scratch | 90.39 / 34.4M (100%) | 90.32 / 65.65M (100%) | 90.28 / 136.96M (100%) |
> | PointNet (scratch) || 89.1 / 3.48M |
> | PointNet (scratch) || 89.2 / 31.32M |
> | PointNet++ (scratch) || 90.7 / 1.48M |
> Table.3. ModelNet 3D Warehouse results (Top-1 accuracy) and trainable parameter size comparison of different ResNet architectures and PointNet series.
>
> We observe that with hundreds of times fewer trainable parameters, FIP-IO+BN achieves competitive or even better performance than training-from-scratch. FIP-IO+BN ResNet50 outperforms the well-known PointNet and PointNet++ with at least 7.78x fewer trainable parameters. Therefore, it is not true that the performance gain is from increasing the number of parameters.
>
> Q5: In 4.2, there is a possibility that the ResNet structure is not a good structure to train on point cloud making it quite bad on scratch training. For pointnet++, the training method is quite strange.
>
> A5:
> We don’t intend to propose or defend any model architectures in this paper. We directly take common architecture with the simplest training recipe to explore our idea. Regarding the ResNet, many well-known point-cloud works are based on this architecture, e.g. PointContrast [3]. It is also noteworthy that as suggested by reviewer KdMv, the proposed image-pretrained model significantly surpasses PointContrast by 0.64 points on the ModelNet object classification.
>
> Regarding PointNet++ [4], it mimics the traditional convolution by sampling centers of the unordered points, grouping their nearby points, and aggregating local features. Therefore, it is straightforward to apply this to images by regarding images as point-clouds. The detailed setting is illustrated in Appendix A.1. We use the standard training recipe and implementation for ImageNet https://github.com/pytorch/examples/blob/master/imagenet/main.py. Interestingly, PointNet++ achieves 31.5% top-1 accuracy and 50.2% top-5 accuracy on the ImageNet validation set, demonstrating that our training strategy guides PointNet++ to learn meaningful image representation. After the image pretraining, we use the regular PointNet++ training recipe to finetune the model for point-cloud recognition.
>
> [1] Quo vadis, action recognition? a new model and the kinetics.
>
> [2] 3-d convolutional encoder-decoder network for low-dose ct via transfer learning from a 2-d trained network.
>
> [3] PointContrast: Unsupervised Pre-training for 3D Point Cloud Understanding.
>
> [4] PointNet++: Deep Hierarchical Feature Learning on Point Sets in a Metric Space.
>
> [5] ImageNet-trained CNNs are biased towards texture; increasing shape bias improves accuracy and robustness.
>
> [6] A computational approach to edge detection.
>
> [7] Bilateral filtering for gray and color images.
>
> [8] Haar wavelet based approach for image compression and quality assessment of compressed image.

---

### Decision · Program_Chairs · 2022-01-20

**Decision:**

Reject

**Comment:**

This paper proposes to transfer the image-pretrained model to a point cloud model by inflating 2D convolutional filters to 3D convolutional filters and finetuning the inflated image-pretrained model, so that 3D point cloud tasks can benefit from 2D image pretraining. Extensive experiments are conducted to validate the effectiveness of the proposed method. Even though the performance gain using the 2D pre-training is notable, the novelty of the paper is limited since inflating 2D model to 3D video action recognition has been studied, and theoretical understanding of the proposed model is lacking. During the rebuttal period, the authors addressed most of the reviewers’ concerns by conducting additional experiments. Even though the performance is compelling, all reviewers agree that the novelty for the paper is limited and the discussion on why this method work is not convincing. Meanwhile, one reviewer points out that some claims made by authors are not well supported. Besides, one reviewer points out that the paper might have a broader impact in a computer vision conference but only provide a limited contribution to the ICLR community. After an internal discussion with reviewers. the AC agrees with the reviewers on their judgments and recommends rejecting the paper because of the limited novelty of the paper.